# Water-Soluble Chemical Vapor Detection Enabled by Doctor-Blade-Coated Macroporous Photonic Crystals

**DOI:** 10.3390/s20195503

**Published:** 2020-09-25

**Authors:** Min-Fang Wu, Hui-Ping Tsai, Chia-Hua Hsieh, Yi-Cheng Lu, Liang-Cheng Pan, Hongta Yang

**Affiliations:** 1Department of Chemical Engineering, National Chung Hsing University, 145 Xingda Road, Taichung City 40227, Taiwan; ssally084411@gmail.com (M.-F.W.); wl02814234@gmail.com (C.-H.H.); initial0210@gmail.com (Y.-C.L.); benny60422@gmail.com (L.-C.P.); 2Department of Civil Engineering, National Chung Hsing University, 145 Xingda Road, Taichung City 40227, Taiwan; huiping.tsai@nchu.edu.tw

**Keywords:** water-soluble chemicals, detecting, doctor-blade coating, photonic crystals, visual colorimetric readout

## Abstract

Water-soluble chemicals, involving a wide range of toxic chemicals in aqueous solutions, remain essential in both daily living or industrial uses. However, most toxicants are evaporated with water through their use and thus cause deleterious effects on the domestic environment and health in humans. Unfortunately, most current low-dose chemical vapor detection technologies are restricted by the use of sophisticated instruments and unable to promptly detect the quantity of diverse toxicants in a single analysis. To address these issues, this study reports the development of simple and fast chemical vapor detection using doctor-blade-coated macroporous poly(2-hydroxyethyl methacrylate)/poly(ethoxylated trimethylolpropane triacrylate) photonic crystals, in which the poly(2-hydroxyethyl methacrylate) has strong affinity to insecticide vapor owing to a favorable Gibbs free energy change for their mixing. The condensation of water-soluble chemical vapor therefore results in a significant reflection peak shift and an obvious color change. The visual colorimetric readout can be further improved by increasing the lattice spacing of the macroporous photonic crystals. Furthermore, the dependence of the reflection peak position on vapor pressure under actual conditions and the reproducibility of vapor detecting are also evaluated in this study.

## 1. Introduction

Commercial water-soluble chemicals, such as cyphenothrin, pyrethrins, pyrethroids, xylenes, and trichloroethane, are capable of dissolving in water and serving as insecticides, pesticides, additives, adhesives, degreaser, and industrial solvents [1]. Nowadays, broad-spectrum aqueous chemical solutions have been widely used in daily living, modern agriculture, and industry, while most poisonous compounds are evaporated with water through their use [2,3]. For instance, water-based insecticides can be directly sprayed onto insects and other surfaces traversed by insects to eliminate disease-carrying insects or to control pests, which damage cultivated plants [4]. However, the insecticide vapors containing even low-dose chemicals lead to serious air pollution and adversely affect both humans and wildlife that are directly exposed to the vapors [5]. According to the report delivered by the National Toxicology Program of the U.S. Environmental Protection Agency, the typical physiological levels of the water-based insecticide are extremely low, in the range of 0.002~0.05 vol.% for cyphenothrin and in the range of 0.005~0.1 vol.% for imiprothrin [6]. Many studies have reported that long-term exposure to the water-soluble chemical vapors can result in dermatitis, anemia, acute lymphocytic leukemia, coronary heart diseases, and even brain tumors [7]. To address these growing issues, developing effective and convenient methodologies for the detection of water vapors containing low-dose water-soluble chemicals is highly demanded.

Recently, numerous reliable technologies, including mass spectrometry, high-performance liquid chromatography, gas chromatography, surface-enhanced Raman scattering, and fluorescence technologies, have been extensively utilized for precisely detecting poisonous compounds [8,9,10,11,12]. Nevertheless, current detection technologies are restricted due to the need for sophisticated instrumentation, a complicated operating procedure, and time-consuming analysis and are difficult to be applied for on-site low-dose chemical vapor detection. In contrast, the detection of low-dose chemical vapor is generally accomplished by various immunoassays and enzyme inhibition assays, which are capable of identifying biological analytes by converting the biological entities into electrical signals [13,14,15,16]. The assays provide comparative advantages in rapid responses, uncomplicated detection processes, and high sensitivity to selected chemicals. Unfortunately, limited availability of antibody, irreversible bondings between antigens and labels, and complex enzyme purification procedures remain challenges in their practical applications [17,18]. In addition, it is even more arduous to detect chemical vapors containing diverse toxicants for most of the existing technologies. Therefore, there is still an urgent demand to provide a straightforward methodology for water-soluble chemical vapor detection.

Photonic crystals comprise periodically arranged structured materials with different refractive indices, generating a forbidden energy gap for electromagnetic waves, and thence allow a certain wavelength range of light to be reflected [19]. Noteworthily, the tunable photonic stopband can be determined by altering the lattice spacing and the effective refractive index of photonic crystals. Owing to the unique optical characteristics of photonic bandgap materials, intense efforts have been devoted to design and develop photonic crystal-based colorimetric sensors [20,21,22]. Condensation of vapors in the cavities within photonic crystals increases the lattice spacing or the effective refractive index of the medium, resulting in a red-shift of reflection peak. The photonic crystals therefore exhibit a corresponding visual readout on exposure to various vapor pressures. Interestingly, the combination of interconnected porous photonic crystals with chemical-responsive polymers can increase the extent of red-shift; hence, the selectivity and sensitivity of chemical detection can be further improved [23,24,25,26,27,28]. However, a great majority of existing photonic crystal-based colorimetric sensors are restricted to detecting one certain chemical and unable to detect water vapors, which contain numerous water-soluble chemicals. Moreover, current lithographic technologies for fabricating photonic crystals are time-consuming and suffer from sophisticated equipment, whereas most self-assembled colloidal methodologies are only accessible for laboratory-scale fabrication [29,30,31,32,33]. It remains a challenge to engineer large-scale photonic crystals for colorimetric detection of water vapors consisting of broad-spectrum chemicals.

Herein, we develop a scalable doctor-blade coating technology for self-assembling macroporous poly(2-hydroxyethyl methacrylate)/poly(ethoxylated trimethylolpropane triacrylate) photonic crystals in a roll-to-roll compatible procedure. A commercial water-based insecticide involving cyphenothrin and imiprothrin is applied to demonstrate the detecting capability. As a result of the poly(2-hydroxyethyl methacrylate) having strong affinity to water and water-soluble chemicals, the as-fabricated photonic crystals rapidly change intrinsic colors upon exposure to water vapors containing water-based chemicals. Furthermore, the color of the macroporous photonic crystals can be recovered after applying a simple water rinsing process. Importantly, the as-constructed photonic crystals are fast-responsive, inexpensive, and portable and exhibit a visual readout for detecting water-soluble chemical vapors.

## 2. Materials and Methods

**Reagents.** The chemicals applied to synthesize silica colloids, containing tetraethyl orthosilicate (98.5 vol.%), absolute ethanol (99.9 vol.%), and ammonium hydroxide (28.5 vol.%), were purchased from Sigma-Aldrich. Deionized water (18.6 MΩ cm) was collected from an HT-15L laboratory deionized water system. Ethoxylated trimethylolpropane triacrylate (ETPTA) monomer, 2-hydroxyethyl methacrylate (HEMA) monomer, ethylene glycol (EG) monomer, and methacrylic acid (MMA) monomer were provided by Sartomer. Initiators 2-hydroxy-2-methyl-1-phenyl-1-propanone (HMPP) and azobisisobutyronitrile (AIBN) were acquired from Ciba-Geigy and Sigma-Aldrich, respectively. Aqueous hydrofluoric acid (48 vol.%) for wet etching silica colloids was purchased from Merck. All reagents were of analytical reagent quality and utilized directly in this research. The commercial water-based insecticide, involving cyphenothrin (0.1 vol.%) and imiprothrin (0.1 vol.%), applied in this research was obtained from SC-Johnson.

**Doctor Blade Coating Technology.** Monodispersed StÖber silica colloids were purified using absolute ethanol to remove any unreacted reagent, followed by dispersing in ETPTA monomers with HMPP (1 vol.%) as a photo-initiator [34]. The silica colloid volume fraction in the suspension was adjusted to be 74 vol.%. The suspension was then doctor-blade-coated uniformly onto a poly(ETPTA) wetting-layer-coated glass substrate. In the coating process, the substrate was dragged at a constant speed (2 cm/min), while the immobilized doctor blade was applied to shear-align the silica colloids. The ETPTA monomers were finally photo-polymerized by exposure to ultraviolet radiation in an X Lite 500 curing system for 3 s to develop a silica colloidal crystal/poly(ETPTA) composite.

**Fabrication of Macroporous Poly(HEMA)/Poly(ETPTA) Photonic Crystals.** The silica colloids embedded in the polymeric composite were wet-etched by immersing in an aqueous hydrofluoric acid solution (2 vol.%) for 5 min to create macroporous poly(ETPTA) photonic crystals. The macroporous film was peeled off from the glass substrate and immersed in an ethanol-based mixture of HEMA monomers (14.5 vol.%), MMA monomers (0.4 vol.%), EG monomers (0.1 vol.%), and AIBN (1 vol.%). The final monomer volume fraction was controlled to be 15 vol.%. In the mixture, MMA monomers and EG monomers can serve as a polymeric toughener and a lubricant, respectively. After eliminating excess mixture by spinning at 1000 rpm for 1 min, the monomers were photo-polymerized by ultraviolet radiation in the curing system for 2 s to engineer a free-standing macroporous poly(HEMA)/poly(ETPTA) photonic crystals.

**Experimental Procedures for Insecticide Vapor Detection.** The as-fabricated macroporous films were placed in a vacuum degassing chamber, which was evacuated and then aerated with demanded insecticide vapor pressure and water vapor pressure. After that, the chamber was inflated with nitrogen to maintain the total pressure at 1 atm. An optical fiber probe sealed in the chamber was utilized to evaluate the normal-incidence optical reflectance from the macroporous films.

**Characterization.** The photographic images and surface morphologies of the macroporous photonic crystals are characterized by a Nikon Z50 digital camera and a JEOL 6335F field-emission scanning electron microscope, respectively. Optical reflection spectra of the macroporous films were assessed using an Ocean Optics HR4000 spectrometer with Ocean Optics DT-MINI-2-GS deuterium tungsten halogen light source and recorded in the wavelength range from 400 nm to 900 nm.

## 3. Results and Discussion

The experimental procedures for engineering macroporous photonic crystal-based insecticide vapor detectors are schematically illustrated in Figure 1. Silica colloidal crystal/poly(ETPTA) composites are self-assembled by a scalable doctor-blade coating technology, whereas the silica colloids are shear-aligned in the coating process, followed by a photo-polymerization procedure [35]. Apparently, the silica colloids, embedded in the poly(ETPTA) matrix, are three-dimensionally and hexagonally close-packed in a long-range ordering (Figure 2). The embedded silica colloids can then be selectively wet-etched using an aqueous hydrofluoric acid solution, while the film can be gently peeled off from the substrate to create free-standing macroporous poly(ETPTA) photonic crystals. As displayed in Figure 3, the macroporous film templated from 355 nm silica colloidal crystals exhibits a shining vermilion color, caused by light reflection from the three-dimensionally ordered macroporous structures. It is noteworthy that the hexagonal close-packed lattice is well retained after the wet etching treatment. The macroporous film can be immersed in an ethanol-based mixture of HEMA monomers, EG monomers, MMA monomers, and AIBN, while the excess monomers are removed in a spin-coating procedure. The monomers are finally photo-polymerized by ultraviolet radiation to engineer a free-standing macroporous poly(HEMA)/poly(ETPTA) photonic crystals.

For investigating the vapor-detecting capabilities of the macroporous poly(ETPTA) photonic crystals, normal-incidence specular reflection spectra of the macroporous film were evaluated under various water vapor pressures at 28 ± 1 °C (Figure 4a). It was found that the normal-incidence reflection peak position of the film was located at 648 nm in a dry nitrogen environment, and thereby the film features a vermilion color. The measured peak position agrees well with the theoretical one (650 nm) estimated by Bragg’s equation:$$\lambda_{peak}=2\;n_{eff\;}d$$
in which $n_{eff}$ and *d* denote the effective refractive index of the medium and the lattice spacing, respectively [36]. This result further demonstrates the long-range ordering of the air cavities in the macroporous photonic crystals. Importantly, water vapor condenses in the cavities, leading to a higher $n_{eff}$ and a lower refractive index contrast between poly(ETPTA) and the enclosed material. As a result, the peak position gradually red-shifts to 788 nm, during which the peak amplitude gradually declines as the water vapor pressure increases from 0 P_0_ to 1.0 P_0_, whereas P_0_ represents the saturation water vapor pressure at 28 °C. Additionally, the changes in the peak position and amplitude were not obvious when the water vapor pressure was higher than 1.0 P_0_. Owing to the fact that the water-based insecticide is composed of a considerable proportion of water, the optical response of the insecticide vapor detecting is similar to that of water vapor detecting (Figure 4b). It was noticed that the reflection peak position shifts from 648 nm to 789 nm, while the corresponding appearance of the film changes from vermilion to colorless as the insecticide vapor pressure reaches 1.0 P_0_ (Figure 4c–f).

To enhance the optical readout of water-based chemical vapor detecting, the macroporous poly(ETPTA) photonic crystals are coated with a thin layer of poly(HEMA), which is responsive to water, to engineer macroporous poly(HEMA)/poly(ETPTA) photonic crystals. The as-developed macroporous film templated from 355 nm silica colloidal crystals displays a striking red color because of light reflection from the highly ordered air cavities (Appendix A). As noticed, the formation of smaller void openings on the macroporous poly(HEMA)/poly(ETPTA) film in comparison with those on the macroporous poly(ETPTA) film demonstrates that a poly(HEMA) layer is uniformly coated. The resulting poly(HEMA) coating layer leads to a higher $n_{eff}$, and a corresponding reflection peak position locates at 680 nm (Figure 5). The poly(HEMA) layer thickness can be calculated using $\lambda_{peak}=2\;n_{eff}d\sin\theta$, in which $n_{eff\;}=VF_{poly\left(ETPTA\right)}\times \;n_{poly\left(ETPTA\right)}+\;VF_{air}\times \;n_{air}+\;VF_{poly\left(HEMA\right)}\times \;n_{poly\left(HEMA\right)\;}$, $n_{poly\left(ETPTA\right)}\;$ equals to 1.44 and $n_{poly\left(HEMA\right)}\;$ equals to 1.45. In addition, $VF_{poly\left(ETPTA\right)}$ and $VF_{air}$ are 0.26 and (0.59$\;-\;VF_{poly\left(HEMA\right)}$), respectively. The estimated volume fraction of poly(HEMA) in the as-fabricated macroporous film is ca. 15.4 vol.%, and therefore a ca. 12.5 nm poly(HEMA) layer is coated on the 355 nm voids. Similarly, the reflection peak position red-shifts to 799 nm and 801 nm under saturated water vapor pressure and saturated insecticide vapor pressure, respectively. As a result, the observed color of the film changes from red to colorless as the insecticide vapor pressure increases from 0 P_0_ to 1.0 P_0_.

Even though the appearance of macroporous poly(HEMA)/poly(ETPTA) film on water-based insecticide vapor detecting was not significantly distinguishable from that of macroporous poly(ETPTA) film, it is clear that the macroporous poly(HEMA)/poly(ETPTA) film exposed to the insecticide vapor presented larger red-shifts (Figure 6a). Moreover, the linear optical response of reflection peak position shift against vapor pressure is crucial in water-soluble chemical vapor detection. The behavior can be interpreted using Flory–Huggins solution theory [37]. Owing to a favorable Gibbs free energy change for mixing poly(HEMA) with water, the poly(HEMA) coating layer features a high degree of swelling while the film is exposed to water-based insecticide vapor. To gain a better understanding of the insecticide vapor detecting capability of the macroporous poly(HEMA)/poly(ETPTA) film, the condensed liquid thicknesses are estimated by Bragg’s equation, wherea $n_{eff\;}=VF_{poly\left(ETPTA\right)}\times \;n_{poly\left(ETPTA\right)}+\;VF_{poly\left(HEMA\right)}\times \;n_{poly\left(HEMA\right)}\;+\;VF_{air}\times \;n_{air}+\;VF_{liquid}\times \;n_{liquid}$, in which $VF_{poly\left(ETPTA\right)}$, $VF_{poly\left(HEMA\right)}$, and $VF_{air}$ are 0.26, 0.15, and (0.59$\;-\;VF_{liquid}$), respectively. By presuming that the vapor condenses on the cavity walls uniformly, the estimated liquid volume fraction ($VF_{liquid}$) can be applied to compute the condensed liquid thickness. As revealed in Figure 6b, a higher vapor pressure is associated with a thicker condensed liquid layer. Additionally, the computed condensed liquid layers of the macroporous poly(HEMA)/poly(ETPTA) film were thicker than those of the macroporous poly(ETPTA) film under various vapor pressures. The deviation is ascribed to the swelling in poly(HEMA) coating layer, resulting in a greater vapor condensation in the macroporous poly(HEMA)/poly(ETPTA) film.

In order to verify the extent of swelling in poly(HEMA) layer, a gravimetric analysis was applied to determine the mass of condensed liquid in the macroporous photonic crystals under various vapor pressures. The mass of condensed liquid could then be utilized to calculate the liquid layer thickness. It was found that the calculated liquid layer thickness using a gravimetric analysis (Appendix A) matched reasonably well with the results as shown in Figure 6b, further confirming the characteristics of the poly(HEMA) coating layer. It is worth mentioning that the poly(HEMA) layer-coated cavities had a smaller space for vapor condensation, and therefore the condensed liquid layers of macroporous poly(HEMA)/poly(ETPTA) film were thinner than those of macroporous poly(ETPTA) film under vapor pressures larger than 1.0 P_0_. To further investigate insecticide vapor detection limit of the macroporous poly(HEMA)/poly(ETPTA) film, diluted commercial water-based insecticides containing 0.01 vol.% chemicals and 0.001 vol.% chemicals were also applied as analytes in this study. As displayed in Appendix A, both of the reflection peak positions red-shifted with the increases of insecticide vapor pressures. The results indicate that the macroporous poly(HEMA)/poly(ETPTA) film has a dynamic range of water-based insecticide detection from 0.1 vol.% to 0.001 vol.%, which is competitive with current vapor sensing technologies [38,39,40]. Importantly, the optical response of the insecticide vapor detecting is indistinguishable from that of water vapor detection for macroporous poly(HEMA)/poly(ETPTA) film. It suggests that low-dose broad-spectrum water-soluble chemical vapors can be detected by employing the as-engineered macroporous photonic crystals.

Since the detection selectivity between water vapor and water-based chemical vapor is not obvious, the water-soluble chemical vapor detecting capability of the macroporous photonic crystals under actual conditions was investigated by evaluating the optical responses under various insecticide vapor pressures in the presence of a fixed water vapor pressure at 0.2 P_0_ (Figure 7). Interestingly, in comparison with insecticide vapor detection in a dry environment, the presence of water vapor brought about larger red-shifts in reflection peaks for either macroporous poly(ETPTA) film or macroporous poly(HEMA)/poly(ETPTA) film. The insecticide vapor condensation, which was accompanied by water vapor condensation in the macroporous film led to a thicker condensed liquid layer and a higher corresponding $n_{eff}$ than those of insecticide vapor detection without water vapor. The characteristics further enhance the red-shift of reflection peak. Importantly, owing to a high degree of swelling on exposure to water, the increased red-shifts of poly(HEMA) layer coated macroporous film could maintain ca. 12.5 nm under various insecticide vapor pressures. The finding is essential for directly identifying water-soluble chemical vapor pressure even under actual conditions. On the other hand, in the presence of insoluble chemicals in water, the condensed non-water-soluble chemicals are incapable of forming a uniform liquid layer on the poly(HEMA)-coated pores. The nonuniform liquid layer brings about light scattering and therefore greatly suppresses the intensity of reflective light.

Lattice spacing is another determining factor in vapor detection. The observed color change of macroporous photonic crystals in insecticide vapor detecting can be easily controlled by adjusting lattice spacing. In this research, macroporous poly(ETPTA) photonic crystals templated from 250 nm silica colloidal crystals were fabricated by as-mentioned doctor-blade-coating technology (Appendix A). The macroporous poly(ETPTA) film displays a shining blue color, originating from light reflection from hexagonal close-packed structures. The reflection peak position of the macroporous film is located at 485 nm and red-shifts with the increase of vapor pressure, resulting in a corresponding color change from blue to yellow (Appendix A). The macroporous poly(ETPTA) photonic crystals can be coated with a poly(HEMA) layer to engineer a macroporous poly(HEMA)/poly(ETPTA) photonic crystals (Appendix A). Similar to previous results, the swelling in poly(HEMA) coating layer in the presence of water led to a thicker condensed liquid layer and therefore facilitated a larger reflection peak shift (Figure 8, Appendix A). The shift of reflection peak is increased under actual conditions. In comparison with the reflection peak position shifts of macroporous poly(HEMA)/poly(ETPTA) films templated from 355 nm silica colloidal crystals (Figure 6a), it is evident that the macroporous poly(HEMA)/poly(ETPTA) films templated from 250 nm silica colloidal crystals displayed smaller red-shifts under different vapor pressures (Appendix A). It is noteworthy that although photonic crystals with a smaller lattice spacing yielded a smaller extent of red-shift, the macroporous poly(HEMA)/poly(ETPTA) film templated from 250 nm silica colloidal crystals featured an evident color changing from blue, green, chartreuse, to yellow under various insecticide vapor pressures. Consequently, the as-engineered macroporous film exhibited a visible readout on water-based chemical vapor detecting.

The requirement of detection time duration plays a critical role in detecting water vapor containing chemicals for practical applications. To investigate the required time duration, the optical responses of the macroporous films templated from 250 nm silica colloidal crystals under fixed vapor pressures were evaluated (Figure 9 and Appendix A). In the presence of 0.5 P_0_ vapor pressure, the reflection peak position of the macroporous poly(ETPTA) film was dynamically altered and remained unchanged after 8 s. Compared with that, the poly(HEMA) coating layer was capable of absorbing condensed water and easily led to the formation of uniform condensed liquid film, resulting from a favorable Gibbs free energy change for mixing poly(HEMA) with water. Therefore, the poly(HEMA)-coated macroporous film allowed a larger red-shift in reflection peak and a shorter detection time duration of 3 s. Importantly, the macroporous poly(HEMA)/poly(ETPTA) film enables an even shorter detection time duration under 0.2 P_0_ vapor pressure, indicating the high efficiency and effectiveness in water-based chemical vapor detection.

It is worthy of attention that the water-based insecticide does not chemically react with poly(HEMA), and thus the condensed insecticide can straightforwardly be removed by water rinsing. Taking advantage of that, the optical properties of the as-engineered macroporous poly(HEMA)/poly(ETPTA) photonic crystals can be wholly recuperated, and the recovered macroporous film can be reused for water-soluble chemical vapor detecting. The reproducibility of insecticide vapor detecting was investigated utilizing a macroporous poly(HEMA)/poly(ETPTA) film templated from 250 nm silica colloidal crystals. The reflection peak red-shifted from 485 nm to 566 nm as the macroporous film was exposed to saturated insecticide vapor and then blue-shifted to 485 nm after a raising treatment. Importantly, the utilization of macroporous poly(ETPTA) scaffold limited inhomogeneous deformation of poly(HEMA) coating layer during the cycles. As shown in Figure 10, the conversion was reversible and repeatable for at least 10 cycles, suggesting that the macroporous photonic crystals provide a reproducible platform for water-based chemical vapor detection. It is worthwhile to note that as compared with the macroporous poly(HEMA)/poly(ETPTA) film templated from 250 nm silica colloidal crystals, the response time and the reproducibility of the film templated from 355 nm silica colloidal crystals did not change. The response time is mainly determined by the Gibbs free energy change for mixing coated polymer with analyte chemicals. Owing to a favorable Gibbs free energy change for mixing poly(HEMA) with water, the poly(HEMA) coating layer is capable of absorbing condensed water and leads promptly to the formation of uniform condensed liquid film. In addition, the use of macroporous poly(ETPTA) as a scaffold presumably limits inhomogeneous deformation of poly(HEMA) hydrogel during the cycles, resulting in a great reversibility for water-based insecticide vapor sensing.

## 4. Conclusions

In summary, a scalable doctor-blade coating technology was developed to engineer macroporous poly(HEMA)/poly(ETPTA) photonic crystals for colorimetric detection of water-soluble chemical vapor. Owing to a high swelling degree of poly(HEMA) in the presence of water and water-soluble chemicals, the macroporous film templated from 250 nm silica colloidal crystals behaves a significant reflection peak shift and a corresponding color change from blue to yellow on exposure to water-based insecticide vapor. The photonic stopband shift can be further improved by increasing lattice spacing of the macroporous photonic crystals. The reflection peak red-shifts with the increase of insecticide vapor pressure even in the presence of water vapor pressure. This suggests that the photonic crystals are capable of evaluating low-dose broad-spectrum water-soluble chemical vapor pressure under actual conditions. Importantly, the use of macrporous poly(ETPTA) scaffold allows a highly reproducible and reversible optical response to water-soluble chemical vapor detection within a very short time. It is believed that the doctor-blade-coated macroporous photonic crystals, coated with various materials with similar solubility parameters to selected chemicals, provide a universal and novel strategy for chemical vapor detections without applying any instrumentation or label.

## Figures and Tables

**Figure 1 sensors-20-05503-f001:**
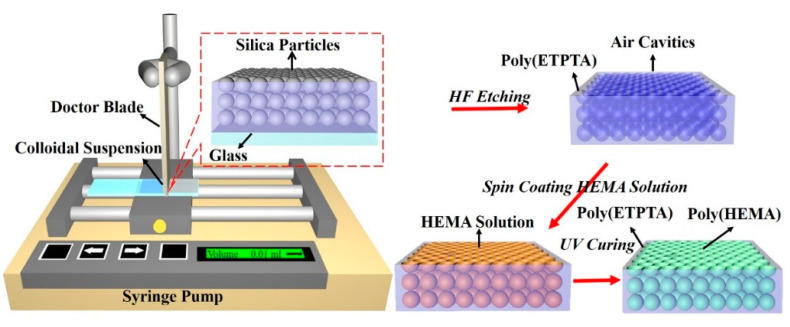
The experimental procedures for self-assembling macroporous photonic crystals.

**Figure 2 sensors-20-05503-f002:**
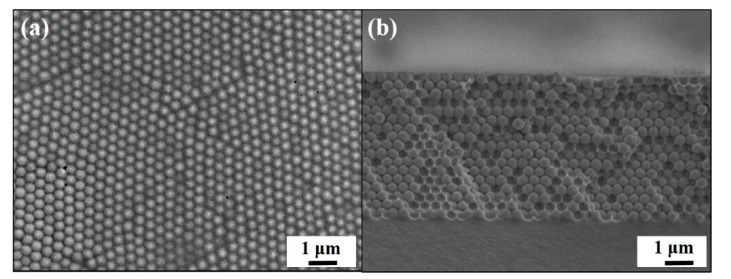
(**a**) Top-view SEM image of a doctor-blade-coated 335 nm silica colloidal crystal/poly(ETPTA) composite. (**b**) Cross-sectional SEM image of the sample in (**a**).

**Figure 3 sensors-20-05503-f003:**
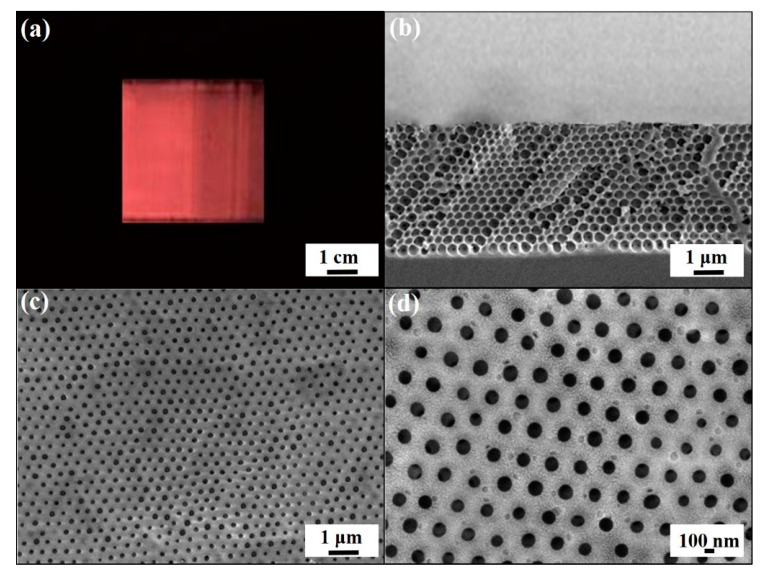
(**a**) Photographic image of a macroporous poly(ETPTA) film templated from 355 nm silica colloidal crystals. (**b**) Cross-sectional SEM image and (**c**) top-view SEM image of the sample in (**a**). (**d**) Magnified SEM image of (**c**).

**Figure 4 sensors-20-05503-f004:**
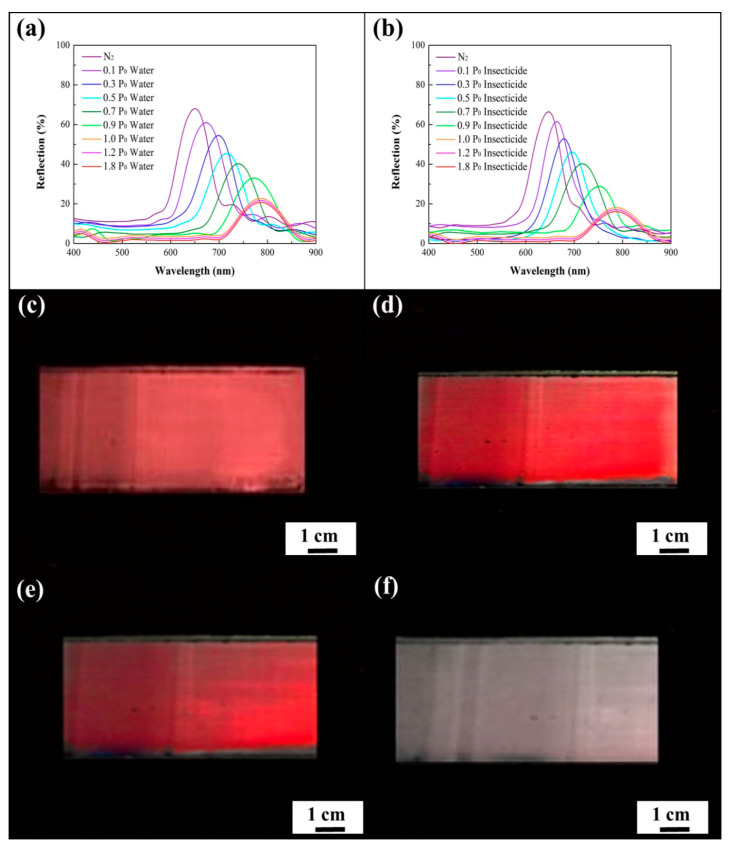
Normal-incidence specular reflection spectra obtained from a macroporous poly(ETPTA) film templated from 355 nm silica colloidal crystals under (**a**) different water vapor pressures and (**b**) different insecticide vapor pressures. Photographic images obtained from the macroporous film under different insecticide vapor pressures. (**c**) 0 P_0_; (**d**) 0.3 P_0_; (**e**) 0.7 P_0_; (**f**) 1.0 P_0_. P_0_ represents the saturation insecticide vapor pressure at 28 °C.

**Figure 5 sensors-20-05503-f005:**
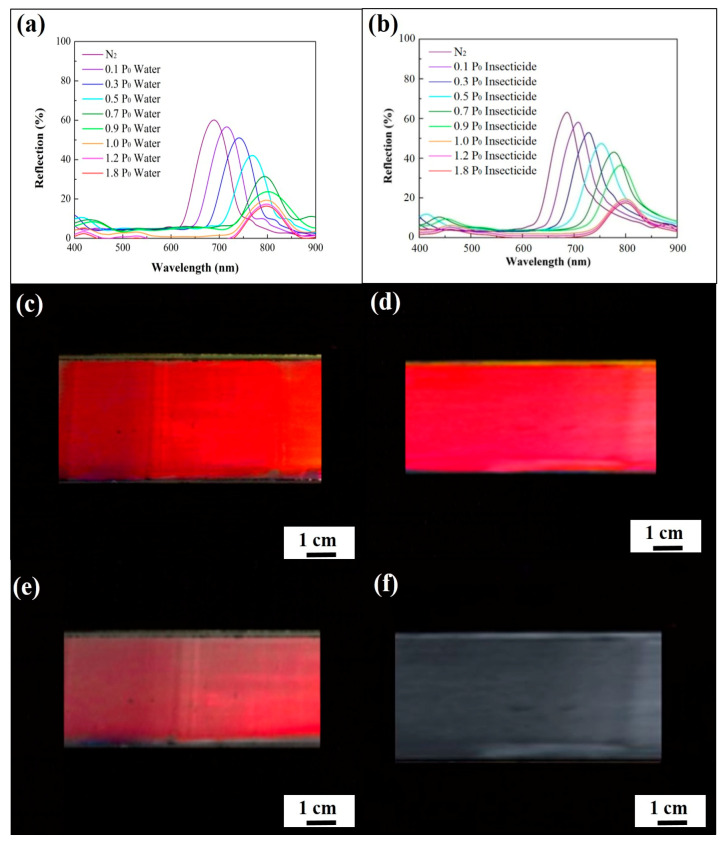
Normal-incidence specular reflection spectra obtained from a macroporous poly(HEMA)/poly(ETPTA) film templated from 355 nm silica colloidal crystals under (**a**) different water vapor pressures and (**b**) different insecticide vapor pressures. Photographic images obtained from the macroporous film under different insecticide vapor pressures. (**c**) 0 P_0_; (**d**) 0.3 P_0_; (**e**) 0.7 P_0_; (**f**) 1.0 P_0_. P_0_ represents the saturation insecticide vapor pressure at 28 °C.

**Figure 6 sensors-20-05503-f006:**
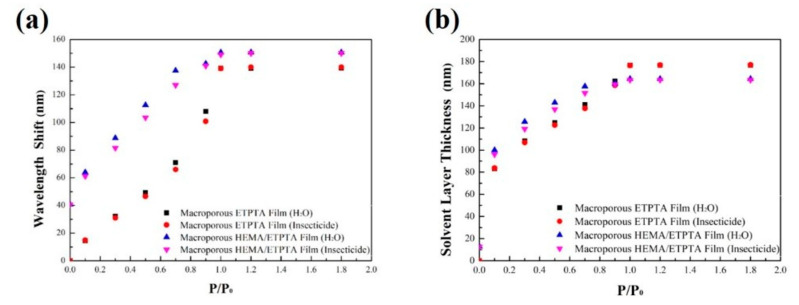
(**a**) The reflection peak position shifts of a macroporous poly(ETPTA) film templated from 355 nm silica colloidal crystals and a macroporous poly(HEMA)/poly(ETPTA) film templated from 355 nm silica colloidal crystals under different vapor pressures. (**b**) Calculated condensed liquid layer thicknesses of the macroporous films under different vapor pressures.

**Figure 7 sensors-20-05503-f007:**
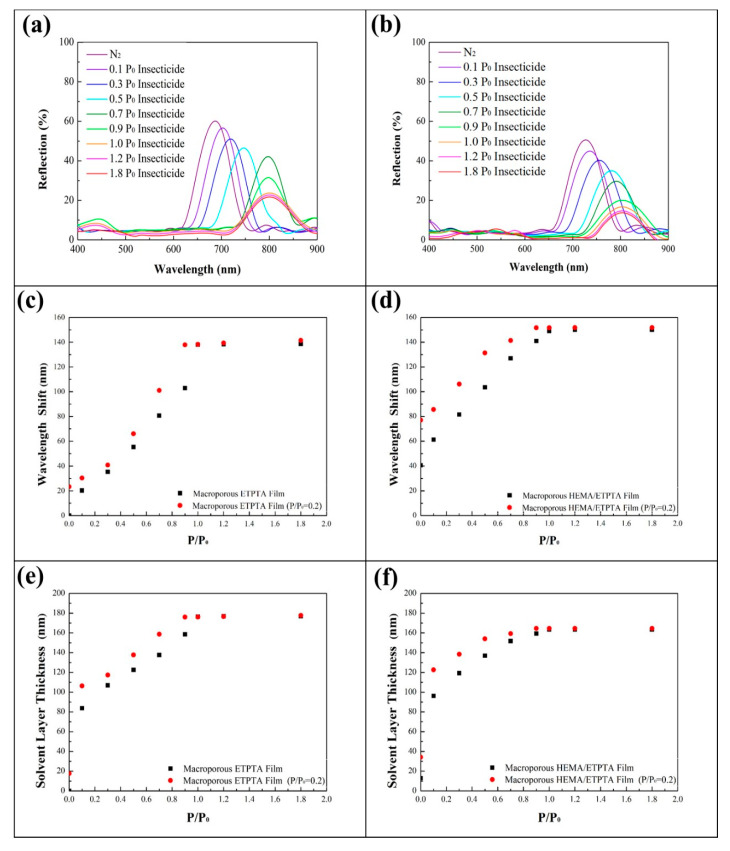
Normal-incidence specular reflection spectra obtained from (**a**) a macroporous poly(ETPTA) film templated from 355 nm silica colloidal crystals and (**b**) a macroporous poly(HEMA)/poly(ETPTA) film templated from 355 nm silica colloidal crystals under a fixed water vapor pressure (0.2 P_0_) and different insecticide vapor pressures. P_0_ represents the saturation water vapor pressure at 28 °C. The corresponding reflection peak position shifts of (**c**) the macroporous poly(ETPTA) film and (**d**) the macroporous poly(HEMA)/poly(ETPTA) film under a fixed water vapor pressure (0.2 P_0_) and different insecticide vapor pressures. The corresponding calculated condensed liquid layer thicknesses of (**e**) the macroporous poly(ETPTA) film and (**f**) the macroporous poly(HEMA)/poly(ETPTA) film under a fixed water vapor pressure (0.2 P_0_) and different insecticide vapor pressures.

**Figure 8 sensors-20-05503-f008:**
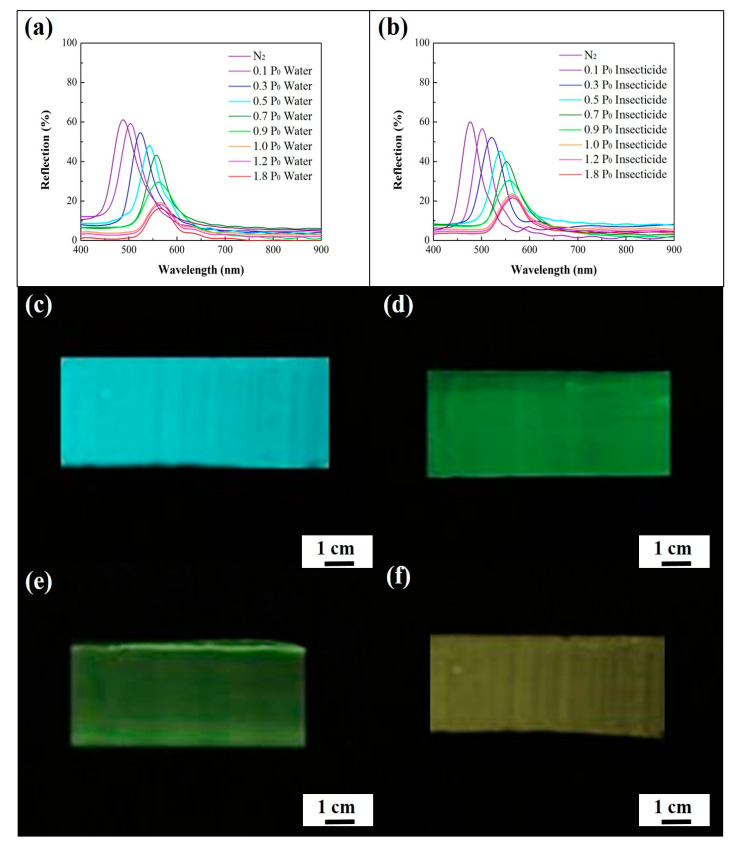
Normal-incidence specular reflection spectra obtained from a macroporous poly(HEMA)/poly(ETPTA) film templated from 250 nm silica colloidal crystals under (**a**) different water vapor pressures and (**b**) different insecticide vapor pressures. Photographic images obtained from the macroporous film under different insecticide vapor pressures. (**c**) 0 P_0_; (**d**) 0.3 P_0_; (**e**) 0.7 P_0_; (**f**) 1.0 P_0_. P_0_ represents the saturation insecticide vapor pressure at 28 °C.

**Figure 9 sensors-20-05503-f009:**
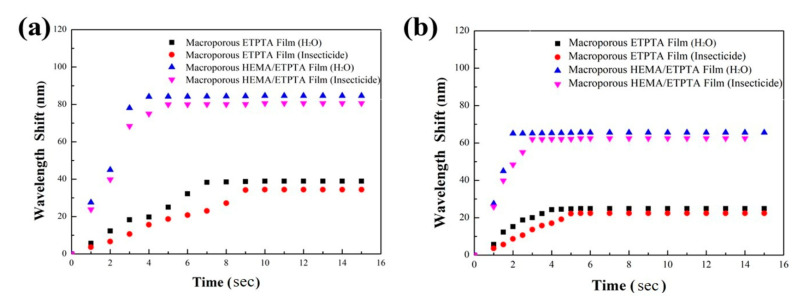
Change of reflection peak position shift with time for a macroporous poly(ETPTA) film table 250 nm silica colloidal crystals and a macroporous poly(HEMA)/poly(ETPTA) film templated from 250 nm silica colloidal crystals under (**a**) 0.5 P_0_ vapor pressure and (**b**) 0.2 P_0_ vapor pressure. P_0_ represents the saturation insecticide vapor pressure at 28 °C.

**Figure 10 sensors-20-05503-f010:**
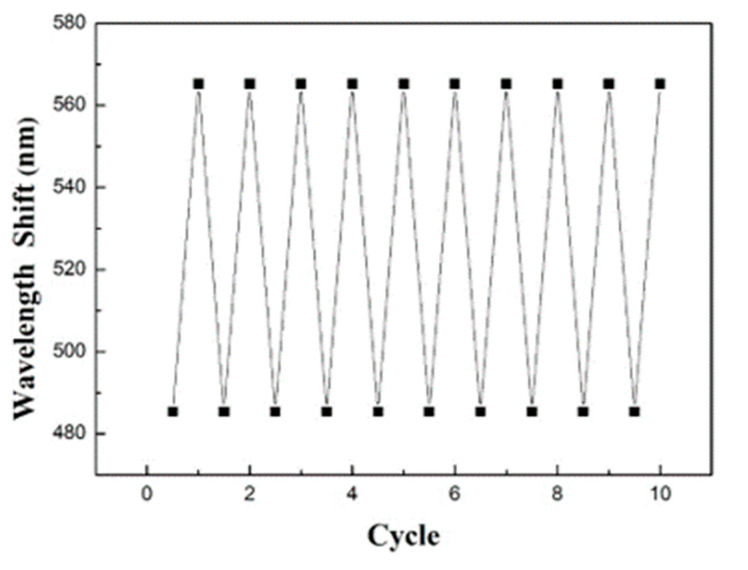
Reproducible insecticide vapor sensing of a macroporous poly(HEMA)/poly(ETPTA) film templated from 250 nm silica colloidal crystals under saturated insecticide vapor.

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
