# Peer review of "Water-Soluble Chemical Vapor Detection Enabled by Doctor-Blade-Coated Macroporous Photonic Crystals"

_sensors, 2020, doi:10.3390/s20195503_

Round 1

Reviewer 1 Report

In this manuscript, Min-Fang Wu et al. report the development of simple and fast chemical vapor detection using doctor blade coated macroporous poly(2-hydroxyethyl methacrylate)/poly(ethoxylated trimethylolpropane triacrylate) photonic crystals, in which the poly(2-hydroxyethyl methacrylate) is with strong affinity to insecticide vapor owing to a favorable. It is believed that the doctor blade coated macroporous photonic crystals, coated with various materials with similar solubility parameters to selected chemicals, provide a universal and novel strategy for chemical vapor detections without applying any instrumentation or label. Thus, this manuscript can be recommended for publication in Sensors after minor revision.

  1. Since the pesticide is proposed in this paper, the detection limit of the toxic substance should be given clearly. The author should also give the test results of different concentrations of toxic reagents.
  2. Why are the titles of the fourth part and the fifth part the same?
  3. The detection technology is very meaningful. Is it universal for the detection of different harmful substances?
  4. Toxic steam will not exist in vacuum environment in real life. Can the author supplement the detection data in air?

Author Response

Please see the attachment. This is the revised version. Thanks. 

Reviewer 2 Report

This research letter deals with the detection of water-soluble chemical vapor with polymer-based photonic crystals. The authors did a pretty good job in the fabrication aspect. However, in the sensor aspect authors reported the ability of the developed photonic crystal (with 355 and 250 nm silica) towards the detection of insecticide vapor detection, the core concept is good but the details and discussion are not appropriate; authors should reform it. The manuscript also floating in between research letter and a full research paper. From my point of view, for the consideration of this manuscript as a research letter, it is too lengthy/lacking in continuity; the authors can shrink the manuscript along with the modifications based on the following suggestions and queries.

  • There are plenty of the problematic water-soluble chemicals exist, the reason for why the authors chose insecticides as an analyte chemical? is not explained well.
  • What is “low-dose chemical”, numerical values should be added and compared with common scale/concentrations and verified with reference techniques.
  • The selectivity of the developed platform is still questionable, there are some possibilities that exist to interfere with the optical signal due to the presence of other compounds present in the water. The authors should strengthen experimentally the selectivity issue. Major aspect.
  • The experimental procedure referred with respect to Figure 1, did not explain completely as there is a lack of continuity.
  • The quality of the pictures and graphs are poor, authors should maintain the image standards with respect to the journal instructions.
  • Line 234. Diameter of 25 nm – how was the value obtained?
  • Figure 9. Longer experimental duration should be evaluated. Some time points in Figure 9b are missing between 5.5-9.5 sec.
  • Authors should improve and reverify the manuscript for the writing errors, especially the referee comment “4. Conclusions” which states his critics on the paper. very unpleasant 
  • The authors should give a dynamic range of detection and its coefficient of determination.
  • The comparison of the reflection shift of the template with 355 and 250 nm silica with respect to the vapor pressure should be discussed in the manuscript.
  • As compared with a scaffold made from 250 nm silica, the response time and the reproducibility of the scaffold made from 355 nm will change? If yes include the comparison also in the manuscript.
  • Figure numbers and captions to be checked and matched with supplementary information also.

Author Response

(The authors gave the same response as above.)

Round 2

Reviewer 2 Report

After careful evaluation of the revised manuscript, I find the corrections and modifications suggested in the previous review report not fully addressed. Hence, there are still a lot of modifications that have to be carried out from the previous review, as it is not suitable to publish with this form. 

In point 2, the authors didn’t provide the answer to “what is low dose chemicals?”

In point 3, the authors' answer is not satisfactory (Instead of the selectivity aspect authors explained the diverged application of the scaffold).

In point 4, and even in the manuscript the usage and purpose of using EG, MMA and ABIN are not clarified.

In point 5, the image resolutions are poor, as noted the reported modifications by the authors are inadequate.

In point 9, the authors did not express the dynamic range clearly.

In point 11, the answer to the query is missing.

In point 12, still all the figures are not explained/indexed properly (mainly images in the grouped caption)

Round 3

Reviewer 2 Report

the authors answered all issues raised for the revised manuscript.